# Comparison of the Formulation, Stability and Biological Effects of Hydrophilic Extracts from Black Elder Flowers (*Sambucus nigra* L.)

**DOI:** 10.3390/pharmaceutics14122831

**Published:** 2022-12-16

**Authors:** Aurelijus Laurutis, Julius Liobikas, Monika Stanciauskaite, Mindaugas Marksa, Kristina Ramanauskiene, Daiva Majiene

**Affiliations:** 1Laboratory of Biochemistry, Neuroscience Institute, Lithuanian University of Health Sciences, LT-50161 Kaunas, Lithuania; 2Department of Biochemistry, Medical Academy, Lithuanian University of Health Sciences, Eiveniu Str. 4, LT-50103 Kaunas, Lithuania; 3Department of Clinical Pharmacy, Faculty of Pharmacy, Lithuanian University of Health Sciences, Sukileliai Avenue 13, LT-50162 Kaunas, Lithuania; 4Department Analytical & Toxicological Chemistry, Faculty of Pharmacy, Lithuanian University of Health Sciences, Sukileliai Avenue 13, LT-50162 Kaunas, Lithuania; 5Department of Drug Technology and Social Pharmacy, Lithuanian University of Health Sciences, Sukileliu Str. 13, LT-50162 Kaunas, Lithuania

**Keywords:** elder flower extract, antioxidant activity, oxidative stress, cell viability, extracellular ROS, intracellular ROS

## Abstract

Elderflower preparations have long been used to treat colds and flu, but their use is undeservedly reduced, and only dried flower teas, less often ethanolic extracts, can be purchased in pharmacies. In the case of homemade teas, the medicinal plant material is extracted with hot water for a relatively short time, thus only a small part of the active substances is extracted. The industrially produced ethanolic extract is rich in active substances, but its use is limited since ethanol in many countries is undesirable and unsuitable for children and geriatric patients. Therefore, the aim of this work was to produce extracts from elder flowers using water as extractant and a mixture of water + polyethylene glycol (PEG) 20%, to compare their chemical composition and stability, and to study the ability to neutralize reactive oxygen species (ROS) and to sustain the viability of C6 glial cells under oxidative stress conditions. The ethanolic extract was used as a standard. Thus, the extract with PEG contained more than two times higher amount of total phenolics (PC) than the aqueous one, and the stability at 6–8 °C was comparable to the stability of ethanolic extract. All three extracts showed an antioxidant effect in a concentration-dependent manner in vitro. However, only the PEG containing extract (at 20–40 µg/mL PC) was the most effective in reducing the intracellular level of ROS and sustaining the viability of glial cells. The results suggest that the co-solvent PEG increases the yield of phenolics in the extract, prolongs the stability, and enhances positive biological effects.

## 1. Introduction

Analyses of the current context of climate change suggest an increasing risk of pathogens arising from hibernation and an expansion of new and already known diseases due to the spread of viruses, bacteria, fungi and protozoans [1]. Infections and infectious diseases are a great burden on many societies. Therefore, to reduce that burden an integrated approach is required, combining health promotion, disease prevention and patient treatment [2]. For instance, antibiotics have been successfully used to fight bacteria-related diseases since 1940; however, the resistance of microorganisms to these preparations is increasing every day [3]. Moreover, antibiotics as well as antifungal drugs, especially when used for a long time, have a series of side effects ranging from allergies to destruction of good bacteria, and even to liver damage [4]. Therefore, herbal preparations known since ancient times could be a good alternative to chemical medications, which usually exhibit many side effects. Black elder (*Sambucus nigra* L.), as one of the sources of natural remedies, has been used for thousands of years as the first medicine for the treatment of flu, cold and respiratory infections [5]. In addition, the literature data support the traditional use of *Sambuci* flos with the following indications: anti-inflammatory, analgetic, mild-diaphoretic, diuretic, expectorant activity, etc. [6]. These plants are widely distributed in the world and are found in the regions of Europe, Asia, North Africa, and North America. Black elder grows by itself in misty and humid places, and it is also cultivated as a medicinal and decorative plant in homesteads. The main medicinal plant raw material of black elder is flowers and fruits [6,7]. Notably, the chemical composition and health properties of black elder fruits have been extensively studied, and nowadays, there is a broad assortment of health products on the market [5]. Therefore, it seems important to reveal the biological and technological properties of elderflowers as a very valuable but relatively undersupplied pharmaceutical raw material.

It has been known for a long time that at the beginning of an illness the tea or infusion made from elderflowers should be used. Currently, scientific studies have shown that black elder flavonoids stop influenza infection by preventing the virus from entering host cells, by competitive inhibition of the virus or by virus endocytosis [1]. In recent times, research has also proven that elderflowers have a wide spectrum of activities including antiviral, antibacterial, as well as immunity stimulating, antioxidant, anticancer, cholesterol-lowering, anti-diabetic and other effects [8]. This raw material has such a wide biological effect due to rich composition of chemical compounds such as flavonoids, phenolic acids, sterols, tannins, vitamins and essential oils [9]. Notably, during the production of extracts, only a part of the biologically active substances passes from the plant raw material to the solvent, and the process is affected by the properties of used solvent. Therefore, it is a matter of great relevance to the technological process to search for and offer the human body environmentally friendly and effective solvents.

Elderflowers can be purchased at the pharmacy in packages of dry plant material, from which one can prepare medicinal tea at home. Using this method, a small amount of active substances is extracted due to the short maceration time. In addition, when poured with hot water, essential oils and other volatile substances evaporate [10]. There are also several types of industrially produced herbal preparations such as syrups and ethanolic extracts. Although syrups are tasty and their 45–60% sugar concentration ensures the stability of solutions, nowadays, when there are many people with diabetes or patients who prefer products without added sugar, the use of syrup as a pharmaceutical form is likely unacceptable [11]. Ethanol extracts are the most widely used in pharmaceutics, as ethanol readily dissolves many active substances in plants [12]. However, due to a relatively high ethanol content: these extracts are not suitable for children and for patients who cannot consume alcohol; they are not recommended in cosmetics; or they are simply prohibited by national laws. Therefore, it is important to search for alternatives to ethanolic solutions for production using hydrophilic solvents or their complexes, which would ensure a good extraction yield, good stability of the produced solutions, and high biological activity.

There is increasing evidence that many diseases are caused by ROS-mediated cell damage. The effect of oxidative stress on the human body is particularly harmful when large amounts of free radicals damage the most important biomolecules of the human body, namely DNA, proteins, and lipids [13]. It is well documented that oxidative stress affects the occurrence and development of diseases such as cancer of various organs, sclerosis, cardiovascular, autoimmune, and neurodegenerative diseases. Additionally, one of the means of prevention and treatment of ailments caused by oxidative stress could be the use of antioxidants. There are several literature sources demonstrating that elderflower aqueous extracts have antioxidant activity [9,14,15]. However, the analyses were mostly performed using only chemical methods, for example, DPPH (1,1-diphenyl-2-picrylhydrazyl), ABTS (2,2′-azino-bis-3-ethylbenzthiazoline-6-sulphonic acid) and other assays [16]. Therefore, it seems relevant to evaluate the antioxidant activity of different elderflower extracts in cells under normal conditions, and to analyze their effect on cell viability under oxidative stress conditions.

Thus, the aim of this work was to produce different liquid hydrophilic extracts from elderflowers, to evaluate their chemical composition and stability, and to compare their properties with ethanolic extract. Moreover, the antioxidant activity of produced extracts was also evaluated and the effects on cell viability by using a glial cell model under oxidative stress conditions were compared.

## 2. Materials and Methods

### 2.1. Chemicals

The raw plant material of *S. nigra* (dry flowers, of a quality equivalent to Ph. Eur. requirements) was obtained from pharmaceutical industry “Emili” (Lithuania) in 2021. Hydrogen peroxide, Folin–Ciocalteu reagent, Dulbecco’s modified Eagle’s medium (DMEM), penicillin-streptomycin solution, trypsin-EDTA solution 0.25% and all HPLC standards were obtained from Sigma-Aldrich Chemie GmbH (Steinheim, Germany). Amplex^®^ Red was purchased from Thermo Fisher Scientific. Cell culture reagents were obtained from Gibco (Fisher Scientific, Waltham, MA, USA). Distilled water was purified using the Milli-Q system (Millipore, Burlington, MA, USA). Ethanol (96%) was manufactured by Vilniaus Degtine (Vilnius, Lithuania). All solvents, reagents, and standards used in this study were of analytical grade.

### 2.2. Preparation of Extracts

A ratio of 1 to 10 (raw plant material to solvent) was used to extract biologically active substances from elder flowers to a solvent. Three solvent systems were used to prepare extracts: water, water-based complex solvent with 20% polyethylene glycol 400 (PEG), and 70% ethanol. Ultrasound-assisted extraction was carried out in an ultrasonic bath (Bandelin Electronic GmbH & Co. KG, Berlin, Germany) for 20 min. The extraction was performed at 20 ± 2 °C temperature and 35 kHz ultrasound frequency. After extraction, solid particles were separated from the liquid by centrifugation at 10,000× *g* for 5 min using Eppendorf centrifuge 5810R. The supernatant was harvested and filtered through a 0.22 µm pore membrane. Prepared extracts were stored in dark vials in a fridge, at 4 °C, until further use.

### 2.3. Cell Line and Cell Culture

Rat C6 glial cell model was chosen as one of the commonest experimental models used for in vitro studies. This cell culture was purchased from the Cell Lines Service GmbH (Eppelheim, Germany). C6 cells were seeded in culture flasks containing DMEM with 10% fetal bovine serum, 100 U/mL penicillin, and 100 µg/mL streptomycin. The cultures were then incubated, at 37 °C, with 5% CO_2_ and saturated humidity. Additionally, 24 h prior to treatment with differently prepared extracts cells were transferred to a 96-well plate at a density of 20,000 cells/well.

### 2.4. Determination of Total Phenolic Content

The method is based on the colorimetric oxidation/reduction reaction using the Folin–Ciocalteu reagent. The used method is based on the general procedure recommended by the European Pharmacopoeia with slight modifications. An amount of 10 µL of each extract was diluted with 1590 µL of purified water and mixed with 100 µL of the Folin–Ciocalteu reagent for 6 min, and later, 300 µL of 20% solution of sodium carbonate were added [17]. The mixture was incubated for 2 h, at room temperature. Absorbance of the solutions was measured at 760 nm wavelength using spectrophotometer Thermo scientific Fluoroskan Ascent. The total phenolic content was expressed in mg of gallic acid equivalents (GAE) per ml of extract.

### 2.5. HPLC Analysis

The predominant phenolic compounds in elderflowers’ extracts were detected by a high performance liquid chromatography (HPLC) using a Waters 2695 chromatographic system with an ACE 5C18 chromatography column (250 × 4.6 mm) and a Waters 996 diode array detector. The obtained data were processed by the Empower 2 Chromatography Data Software. HPLC eluents consisted of 0.1% trifluoroacetic acid (eluent A) and 100% acetonitrile (eluent B). The elution program was used as follows: from 5% to 15% eluent B at 0–8 min, from 15% to 20% eluent B at 8–30 min, from 20% to 40% eluent B at 30–48 min, from 40% to 50% eluent B at 48–58 min, from 50% to 50% eluent B at 58–65 min, from 50% to 95% eluent B at 65–66 min, from 95% to 95% eluent B at 66–70 min, and from 95% to 5% eluent B at 70–71 min. The mobile phase flow rate was 1 mL/min, and the total flow time 81 min. The injection volume of extract was 10 µL. The column temperature was 25 °C. Compounds present in the samples were identified by the UV absorption at a wavelength range of 300–380 nm and by the retention time of analytes and reference substances [18]. The reference substances: chlorogenic acid (R^2^ = 0.9999), neochlorogenic acid (R^2^ = 0.9999), rutin (R^2^ = 0.9999), isoquercitrin (R^2^ = 0.9999), and quercetin (R^2^ = 0.9999).

### 2.6. Evaluation of Extracellular Hydrogen Peroxide

The H_2_O_2_ amount was determined fluorometrically using 10-acetyl-3,7-dihydroxyphenoxazine (Amplex Red). In combination with horseradish peroxidase, this dye reacts with H_2_O_2_ in a 1:1 stoichiometry to produce the red-fluorescent resorufin [12]. Hanks’ Balanced Salt Solution (HBSS) was enriched with 100 µM of H_2_O_2_ and different amounts (0.5–40 µg/mL PC) of investigated extracts. After that mixtures of wells were subjected to Amplex^®^ Red (5 µM) in the presence of horseradish peroxidase (HRP; 2 U/mL). The fluorescence intensity of the resulting resorufin was detected by a fluorometer (Ascent Fluoroskan, Thermo Fisher Scientific, Inc., Waltham, MA USA) at excitation and emission wavelengths of 544 and 590 nm, respectively. For the control, the level of H_2_O_2_ was determined in HBSS containing the appropriate amount of the solvents only.

### 2.7. Measurement of Intracellular Reactive Species

Intracellular ROS were assessed using the 2,7-dichlorofluorescein diacetate (DCFH-DA) [12]. After the incubation of C6 cells in 96-well plates for 24 h, they were incubated with DCFH-DA (10 µM) in HBSS, at 37 °C, for 30 min. During this time, a part of DCFH was diffused into the cells. The excess dye was washed twice with phosphate-buffered saline. Wells were filled with an HBSS and different amounts (0.5–40 µg/mL PC) of investigated extracts were added. In the presence of cellular oxidizing agents, DCFH is oxidized to the highly fluorescent compound dichlorofluorescein, so the fluorescence intensity is proportional to the amount of ROS produced in the cells. The fluorescence of dichlorofluorescein was detected by a fluorometer at excitation and emission wavelengths of 488 and 525 nm, respectively. The control level of intracellular ROS was determined using appropriate amounts of solvents and the fluorescence intensity in control samples at time point “0” was assumed to be 100%.

### 2.8. Assessment of Cell Viability

To investigate the effect of extracts on cell viability in oxidative stress conditions cells were treated: (1) high oxidative stress conditions (100 µM H_2_O_2_ for 24 h with different amounts of investigated extracts) or (2) moderate oxidative stress conditions (75 µM H_2_O_2_ for 6 h with different amounts of investigated extracts). After the treatments, viability staining of nuclei was performed by adding propidium iodide (PI, 3 µg/mL) and Hoechst 33,342 (6 µg/mL) to the incubation media and allowing the dyes to penetrate cells, incubating for 5 min, at 37 °C. The stained cells in the cultures were visualized under a fluorescent microscope OLYMPUS IX71SIF-3 (Olympus Corporation, Tokyo, Japan). Hoechst33342-only-positive nuclei exhibiting blue fluorescence were considered viable, and Hoechst3334-plus-PI-positive nuclei stained magenta were identified as necrotic. The image analysis was performed using ImageJ software.

### 2.9. Statistical Analysis

Results are presented as the means of 3–7 experiments (performed in three technical replicates) ± standard error. Statistical analysis was performed by one-way analysis of variance (ANOVA), followed by Dunnett’s post-test using the software package SigmaPlot version 13.0 (Systat Software Inc., Slough, UK). The value of *p*-value < 0.05 was taken as the level of significance.

## 3. Results

### 3.1. Chemical Composition of Water, Polyethylene Glycol-Water and Ethanolic Extracts from Elder Flowers

The aim of this experiment was to compare chemical composition of extracts prepared with different solvents: water (WES); water with 20% co-solvent PEG (Pg-WES); and 70% ethanol (EES), which was used as a reference extract. The chemical composition of investigated extracts were determined by HPLC method, whereas the total content of phenolic compounds (PC) was obtained by the Folin–Ciocalteu method, and the results are presented in Table 1.

Thus, the lowest amount of selected phenolic compounds, as well as the total content of PC were found in WES. In the context of the results obtained with the hydrophilic solutions, it is worth noting that the addition of PEG co-solvent to water led to a more than two-fold increase in the total PC. Moreover, the amount of selected phenolic compounds was also higher in Pg-WES then in WES, and close to the levels of phenolic compounds found in EES.

### 3.2. Stability of Differently Prepared Extracts from Elder Flowers

The process of design and production of pharmaceutical products possessing health–improving activities must consider the stability of preparations and optimal storage conditions. Therefore, the aim of the following experiment was to evaluate the stability of three different extracts for 6 months, at room/ambient temperature (20 ± 2 °C), or in a refrigerator (6–8 °C). The results are presented in Table 2.

Thus, the organoleptic properties of WES stored at room temperature changed after the first month, but the total content of PC did not change. After 3 months, the solution became turbid, and the study was discontinued. However, the organoleptic properties of WES, which was stored at 6–8 °C, slightly changed after 3 months of incubation. The total amount of PC, as compared to the initial time point, statistically significantly decreased only after 6 months of incubation. Notably, the total amount of PC in the Pg-WES extract did not change statistically significantly over the studied period, irrespective of storage temperature. However, the extract stored at room temperature became turbid after 6 months. In contrast, the EES did not show any decrease in the amount of total PC throughout the study, nor did it demonstrate any change in organoleptic properties either at room temperature or after incubation in the refrigerator.

Thus, it could be suggested that the addition of PEG as a co-solvent to the aqueous solution resulted in retained organoleptic properties of the extract and the unchanged total content of PC after 6 months storage of Pg-WES at 6–8 °C, as compared to the WES.

### 3.3. The Effect of Differently Prepared Extracts from Elder Flowers on Extracellular ROS Concentration

The aim of this experiment was to reveal and compare the effectiveness of different extracts in neutralizing reactive oxygen species (ROS), namely, hydrogen peroxide, in cell culture medium. Thus, the cell culture medium was supplemented with 100 µM of H_2_O_2_ and the antioxidant activity of rutin and different extracts at various concentrations (expressed as µg of PC per ml) was analyzed.

It was found that a statistically significant effect (14%) of rutin was observed already at 5 µg/mL (Figure 1a), and the strongest effect (70–87%) was reached at 25–40 µg/mL. The WES extract demonstrated a similar pattern of efficiency in decreasing the level of hydrogen peroxide in cell culture medium (Figure 1b). Thus, the level of hydrogen peroxide was decreased by 20% at 5 µg/mL PC, and reached 11–14% of the initial level of hydrogen peroxide at 35–40 µg/mL PC.

Notably, the Pg-WES extract (Figure 1c), was also effective (by 12%) starting just at 5 µg/mL PC. Moreover, at its highest concentrations (35–40 µg/mL PC) the extract reduced the level of hydrogen peroxide down to 8–10% of the initial concentration. This result is comparable to the effects of EES on the levels of hydrogen peroxide (10–15% at 35–40 µg/mL PC) (see Figure 1d). However, the EES extract was found to be effective even at 1 µg/mL PC and reduced the level of hydrogen peroxide by 15%.

### 3.4. The Effect of Elderflower Extracts on the Intracellular ROS Concentration in a Glial Cell Culture

It is well documented that intracellular ROS can damage proteins, membrane lipids and even RNA and DNA molecules, and also, they can impair cellular functions and lead to cell death [13]. Therefore, this experiment aimed at the determination of intracellular ROS levels after cell incubation with rutin or different elderflower extracts.

Rutin was found to increase a level of intracellular ROS by 10% at 20 µg/mL after 2 h of incubation with cells (Figure 2a). Moreover, the level of ROS further increased by 9% and 15% at much higher concentration (40 µg/mL) after 1.5 h and 2 h of incubation, respectively, whereas the WES at 10–20 µg/mL PC decreased the intracellular level of ROS by 15–19% (Figure 2b). The level of ROS further decreased by 20–23% at 40 µg/mL PC after 1.5 and 2 h of incubation, respectively.

Notably, the Pg-WES was found to be effective at much lower concentrations (Figure 2c). The extract at 5–10 µg/mL PC statistically significantly reduced the level of intracellular ROS by 9–15% after 1.5 and 2 h. In addition, the level of ROS decreased even more: by 29–32% after 1.5 h and 32–36% after 2 h of incubation with 20 and 40 µg/mL PC, respectively.

It is worth noting that EES already at 1–10 µg/mL statistically significantly reduced the level of ROS by 10–17% after 1.5–2 h of incubation (Figure 2d). At 20–40 µg/mL PC the level of intracellular ROS decreased at 0.5–1 h of incubation. However, the ROS level was not statistically significantly different from control after 1.5 and 2 h of incubation.

### 3.5. The Effect of Elderflower Extracts on Glial Cell Viability under Oxidative Stress Conditions

In previous experiments we have demonstrated that all extracts possess extracellular ROS neutralizing activity and some of them decrease intracellular ROS level in vitro. Therefore, it could be hypothesized that those extracts could also demonstrate a protective effect on cells under oxidative stress conditions.

#### 3.5.1. High Oxidative Stress Conditions

At first the effect of high oxidative stress (100 µM H_2_O_2_ for 24 h) on glial cell culture was evaluated. It was found that after incubation of cells in these experimental conditions all cells underwent necrosis (Figure 3b) The presence of WES at 10 µg/mL PC did not reduce the damaging effect of 100 µM H_2_O_2_ on cell viability (Figure 3f). However, the number of viable cells increased in wells treated with 20 µg/mL PC, and further reached 23 ± 4% of viable cells in the presence of 40 µg/mL PC (Figure 3c,f). In contrast, the Pg-WES effectively increased the number of viable glial cells at its lowest concentration—10 µg/mL PC. A stronger and statistically significant positive effect (27 ± 4% of viable cells) was observed at 20 µg/mL PC, and the proportion of viable cells outnumbered the dead ones at the highest tested (40 µg/mL PC) concentration: 52 ± 5% vs. 48 ± 4% (Figure 3d,f). Notably, the EES did not have any statistically significant effect on the viability of glial cells in the presence of 100 µM H_2_O_2_ (see Figure 3e,f)

#### 3.5.2. Moderate Oxidative Stress Conditions

Next the glial cells were incubated with 75 µM H_2_O_2_ for 6 h, i.e., under relatively moderate oxidative stress conditions. Thus, this treatment resulted in the appearance of higher population of dying cells (approx. 9%) when compared to the control (Figure 4a,b). Moreover, the WES and Pg-WES extracts at their lowest concentrations (1–5 µg/mL PC) did not affect the viability of cells after treatment with H_2_O_2_. However, at much higher concentrations (10 µg/mL PC and above) the WES and Pg-EES extracts restored the number of viable cells to the control level (Figure 4c,d).

A completely different picture was observed after treatment with the EES extract (see Figure 4e). At high concentrations (10 µg/mL PC) the extract statistically significantly reduced the number of viable cells. Furthermore, a statistically significant deleterious effect was also observed at 5 µg/mL PC and even at 0.5 µg/mL PC, and the level of dying cells was close to the one observed in wells with 75 µM H_2_O_2_: 14% and 11%, respectively. Notably, the EES extract only at 1–2.5 µg/mL PC statistically significantly reduced the number of dying cells and consequently increased the number of viable cells to the control level.

## 4. Discussion

The medicinal raw material of the plant used in this work is the flowers of black elder. According to the European Pharmacopoeia (Ph. Eur. 01/2008:1217) the elderflowers are described as having a strong, aromatic and characteristic smell and a sweet, slightly bitter taste. After preparing of extracts from elderflowers and evaluating of the total PC, the lowest amount of active compounds was found in the aqueous extract (see Table 1). However, this amount was about 10 times higher compared to the results of other scientists who produced the tea infusions, at room temperature, for 15 min (total PC from 15.23 to 35.57 mg GAE/g dry weight of elderflowers [19]. The ethanolic extract of elderflowers was produced for comparison, which also contained a higher amount of extracted compounds than the results published by other researchers: total PC from 13.6 to 27.19 mg GAE/100 mL in home prepared tinctures of elderflowers [20]. Undoubtedly, a higher yield of active compounds in our extracts was achieved by ultrasound-assisted extraction, as cavitation can result in the breakage of cell walls, thus accelerating the release of cellular content and promoting the penetration of solvent into the damaged cells [21]. Although more active substances were extracted from elderflowers by using ethanol as a solvent, we aimed at offering the most suitable solvent for all patient groups, including children, and as the most friendly candidate for cosmetic and pharmaceutical products.

Polyethylene glycol (PEG-400) is known to be readily miscible with water and significantly improves the water solubility of polyphenolic compounds found in plants and other poorly soluble pharmaceutical preparations [22]. The use of this co-solvent in the production of various aqueous extracts is gaining popularity no only due to its proven ability to enhance the extraction of poorly soluble active substances, but also due to its antibacterial activity against various pathogenic bacteria, including *Klebsiella pneumoniae*, *Pseudomonas aeruginosa*, *Escherichia coli* and *Staphylococcus aureus* [23]. Thus, the extract produced with this co-solvent contained a statistically significantly higher level of active compounds compared to WES but lower compared to EES. In addition, Polyethylene glycol has also contributed to a higher stability of extract when compared to the stability of aqueous extract without PEG. Such an effect could be attributed to the antimicrobial activity of PEG. Moreover, the stability of Pg-WES extract was very similar to that of the EES. Notably, pharmaceutical manufacturing allows the addition of up to 30% PEG, which would likely further increase active ingredient extraction and solution stability. However, due to the optimal viscosity properties and proper bioassay performance, the 20% PEG solution was chosen for our experiments [24].

Elderflower extracts are known to contain several main compounds, namely, rutin, protocatechuic acid, chlorogenic acid, neochlorogenic acid, kaempferol, and quercetin [5,25]. It is worth noting that the results of HPLC analysis of our extracts are in line with the data published in other studies and show that the main active substances are flavonoids (rutin being one of the most predominant) and phenolic acids. Notably, many studies have already shown that these substances have a multifunctional effect, including an antioxidant one [15]. The antioxidant potential is directly related to the radical scavenging ability. Therefore, in a first series of bioactivity studies, we determined the ability of extracts to neutralize ROS in the cell culture medium. It is known that inside the cell, the main producer of ROS is the mitochondria, which at different points in the respiratory chain can release superoxide radicals into the environment [26]. Subsequently, the radicals are rapidly converted by superoxidases into H_2_O_2_. This non-radical molecule is quite stable, and it easily diffuses outside the cell wall to the extracellular space. H_2_O_2_ is therefore accepted as the main substance with oxidative activity in the extracellular space [27]. The concentration of this substance used in our experiments was chosen based on the results from studies that measured H_2_O_2_ concentration during oxidative stress [28]. Thus, we found that the lowest concentration of total PC (0.5 μg/mL PC) of all extracts (see Figure 1b–d), did not affect the concentration of H_2_O_2_ in the cell culture medium. However, the level of H_2_O_2_ in the cell culture medium was decreasing with increasing concentration of total PC in all tested extracts. The strongest effect (by 87–92%) on the level of H_2_O_2_ was observed with the highest concentration of total PC (40 μg/mL) used in our study. Notably, the EES extract was the most effective among tested extracts at the lower levels of total PC. For instance, a 50% reduction in the level of H_2_O_2_ was already achieved at 10 µg/mL PC from EES, whereas the WES and Pg-WES extracts reached a similar effectiveness at about 15 µg/mL PC. It is also worth noting that rutin, one of the major compounds found in elderflower extracts, had a weaker effect on the level of H_2_O_2_ in the cell culture medium if compared to the effects of extracts (see Figure 1a). Thus, at the highest used concentration (40 μg/mL) rutin reduced the level of H_2_O_2_ by 70% as compared with control. In contrast, a similar effect by extracts was reached at much lower concentrations of PC: at 25 µg/mL PC from WES, 20 µg/mL PC from Pg-WES, and 15 µg/mL PC from EES. Thus, the extracts containing lower concentrations of rutin in combination with other phenolic compounds achieved a stronger reduction in H_2_O_2_ level in the cell culture medium than rutin alone. No data were found in the literature to compare the effects of flower extracts on extracellular ROS, but elder leaves extracts were used under similar experimental conditions: the elder *Sambucus ebulus* L. leaves extract (with 50% of ethanol) at 50 μg/mL PC reduced the level of H_2_O_2_ by 64% if compared to control [7]. Thus, our results, as well as those of other group of scientists indicated that the phenolic compounds of elder extracts in the range of 1–50 μg/mL PC effectively neutralize extracellular ROS.

The effect of the extracts on intracellular radical generation was also analyzed. There are data showing that rutin at micromolar range (20 and 40 µM) decrease intracellular ROS production in bone marrow-derived macrophages [29]. Another group of researchers studying human neuroblastoma cells (IMR32) in vitro found that high (100 µM) and low (100 nM and 10 µM) rutin concentrations significantly avert ROS generation after 24 h of rutin exposure [30]. However, in our experimental model, rutin used at 0.5–10 μg/mL had no effect during the 2 h period, whereas higher concentrations (20–40 μg/mL) even increased ROS levels by 10–15%. EES used at similar concentrations (20–40 μg/mL PC), although tending to decrease at the beginning of the study, did not show any difference in ROS levels from the control samples after 1.5 and 2 h. However, the extract used in small amounts (1–15 µg/mL PC) showed antioxidant activity. As EES is made with 70% ethanol, it contains water-soluble substances, but mainly water-insoluble substances, which, when introduced into the cell culture medium, form a microsuspension that is deposited as fine insoluble particles on the bottom of the plate with cells. Similarly, the addition of a solution of rutin, which is soluble in ethanol, to the cell culture medium results in the formation of a microsuspension due to the change in solvents, which does not exhibit antioxidant activity in this cell model. In contrast, the results of the hydrophilic extracts showed that WES and Pg-WES exhibited a concentration-dependent ROS level-decreasing effect (Figure 2b,c). The lowest concentration of WES that had a statistically significant ROS level-reducing effect after 2 h was 10 μg/mL, whereas the effect of Pg-WES was already observed at 5 μg/mL. The highest concentration of WES used (40 μg/mL) had a 23% reduction in ROS level after 2 h, whereas the same concentration of Pg-WES under similar conditions had a 32% reduction in ROS level. These results are in accordance with other authors, who have studied the effects of elder flower extract on the level of intracellurlar ROS. Palomino and co-workers showed that 25 and 50 µg/mL of aqueous and ethanolic extracts significantly reduced the steady-state concentration of ROS, indicating that the amount of phenolic compounds in both doses was enough to decrease the basal ROS production in cultured SH-SY5Y cell line [31].

Oxidative stress is a common problem in modern life, as there are many internal (bacterial and viral infections, inflammation, stress, etc.) and external factors in the body that contribute to ROS generation [13]. In our experimental model, firstly, we created conditions of high oxidative stress (100 µM H_2_O_2_, 24 h). This treatment resulted in the death of all cells. The same result was found if cells under oxidative stress conditions were additionally treated with ethanolic elderflower extract. In contrast, a concentration-dependent increase in cell survival was found for the treatment using extracts made with hydrophilic solvents. Although 10 µg/mL WES had no effect, the number of viable cells increased with increasing concentrations of the extract, and a statistically significant number of viable cells (23%) was found at the highest WES concentration tested in these experiments, 40 µg/mL PC. Even better results were obtained with Pg-WES, where viable cells were already found at 10 and 20 µg/mL, and at 40 µg/mL, the number of dead cells was halved. It is likely that the effect of this extract being better compared to the others is due to the property of the co-solvent PEG which increases the penetration of the active substances. The results of Ma and co-workers also demonstrate that PEG400 could act as more than a co-solvent, as it is an effective chemical permeation enhancer [22].

Having evaluated the effects of the extracts under conditions of high oxidative stress, we wanted to investigate their effects under the milder conditions that organisms are likely to face in everyday life. Under the selected conditions (75 µM H_2_O_2_, 6 h) only 9% of the cells were undergoing necrosis. Under these conditions, both hydrophilic extracts used at concentrations of 10 µg/mL and above reduced cell death to control levels, although the lower concentrations tested (1 and 5 µg/mL) did not have this effect. In contrast, EES was effective in reducing cell death at low concentrations (1–5 μg/mL), whereas the higher concentrations of this extract tested did not have a cell viability protective effect.

## 5. Conclusions

The addition of 20% co-solvent PEG-400 to the aqueous solution increased the extraction of phenolic compounds from black elder flowers by more than two-fold, and increased the yield of the main active substances such as quercetin and its derivatives and phenolic acids. However, the PEG-containing solvent did not have the same extraction power as ethanol.

All extracts reduced the level of H_2_O_2_ in cell culture medium in a concentration-dependent manner. Notably, the Pg-WES was the most effective in reducing the level of intracellularly generated ROS in vitro, while the WES demonstrated a lower, albeit statistically significant, effectiveness. The EES showed its antioxidant activity at low concentrations.

Under high oxidative stress, when exposure to 100 µM H_2_O_2_ for 24 h resulted in the death of all cells, the Pg-WES had the highest protective effect, since it reduced the population of dying cells by half when applied at 40 µg/mL PC, whereas the WES at the same concentration was half as effective. At milder oxidative stress conditions (75 µM H_2_O_2_ for 6 h and only 9% of cell death), both hydrophilic extracts (starting at 20 µg/mL) restored the cell viability to the control levels. The EES, used at 1–5 μg/mL PC, had a comparable effect.

Therefore, the obtained results suggest that PEG is a potent co-solvent, since it increases the yield of phenolic compounds in aqueous extract, prolongs its stability, and enhances positive biological effects.

## Figures and Tables

**Figure 1 pharmaceutics-14-02831-f001:**
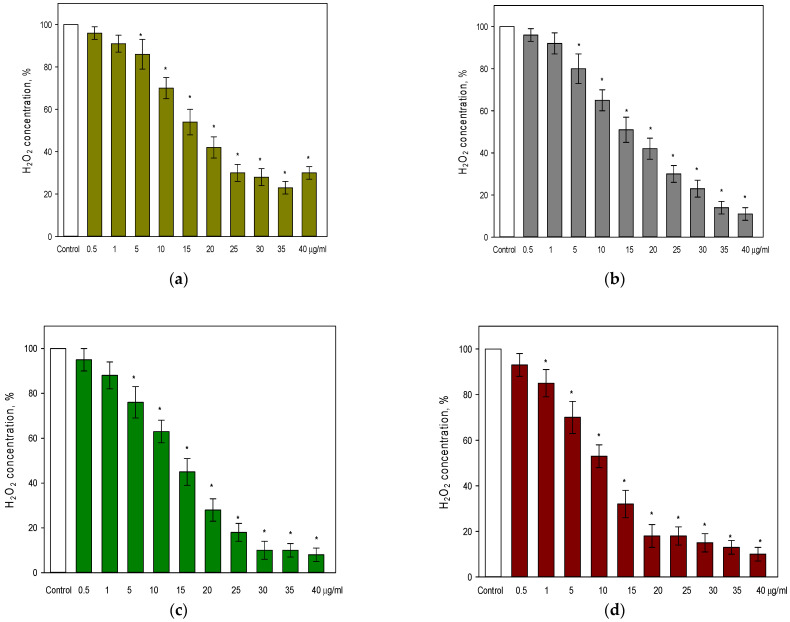
Hydrogen peroxide levels in cell culture medium after incubation with rutin and elderflower extracts: (**a**) rutin; (**b**) WES extract; (**c**) Pg-WES extract; (**d**) EES extract. For details, see the Materials and Methods section. * statistically significant difference vs. control.

**Figure 2 pharmaceutics-14-02831-f002:**
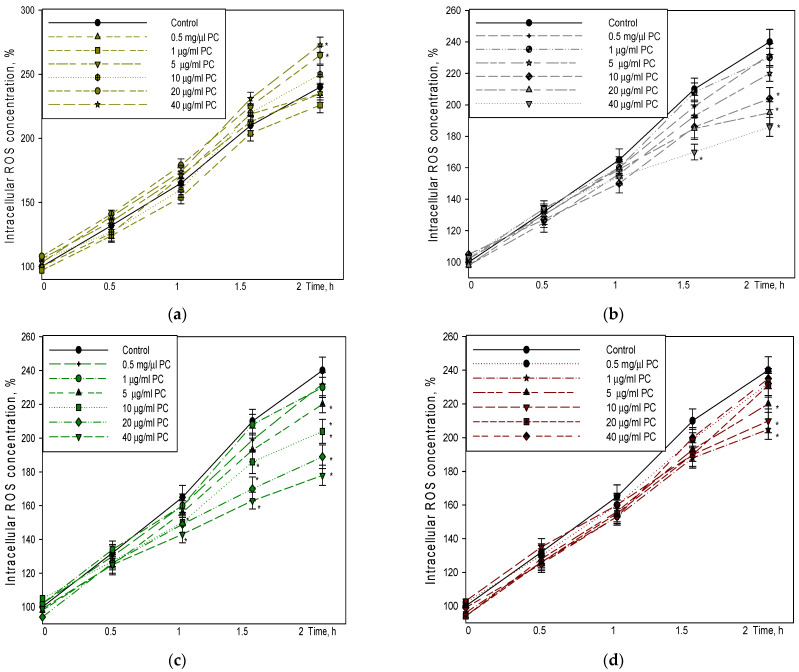
Effect of rutin and elderflower extracts on the intracellular level of ROS: (**a**) rutin; (**b**) WES extract; (**c**) Pg-WES extract; (**d**) EES extract. Cells were incubated for 30 min with 10 μM DCFH-DA in HBSS and with or without rutin or different extracts for 0, 0,5, 1, 1.5 and 2 h. For details, see the Materials and Methods section. * statistically significant difference vs. control. Missing error bars are smaller than the symbol size.

**Figure 3 pharmaceutics-14-02831-f003:**
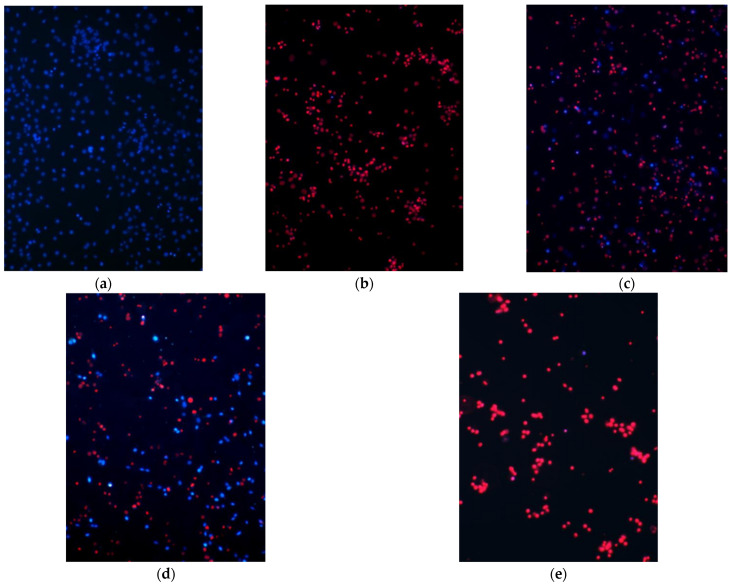
Effectivity of differently prepared extracts from elderflowers in sustaining the viability of glial cells under oxidative stress conditions. Cells were incubated for 24 h with 100 µM H_2_O_2_ or appropriate solvent (control) and additionally in the presence of various concentrations of the elderflower extracts. At the end of experiment, the cells were double-stained with Hoechst 33,342 and PI, and the viability was assessed under fluorescence microscope. Original magnification ×20. For details, see the Materials and Methods section. Typical photographs of cells after treatment with (**a**) Control, (**b**) 100 µM H_2_O_2_, (**c**) 100 µM H_2_O_2_ and 40 µg/mL PC of WES, (**d**) 100 µM H_2_O_2_ and 40 µg/mL PC of Pg-WES, (**e**) 100 µM H_2_O_2_ and 40 µg/mL PC EES, whereas (**f**) cell viability (%) after treatment with H_2_O_2_ in the presence of WES, Pg-WES and EES. * statistically significant difference vs. H_2_O_2_ 100 µM.

**Figure 4 pharmaceutics-14-02831-f004:**
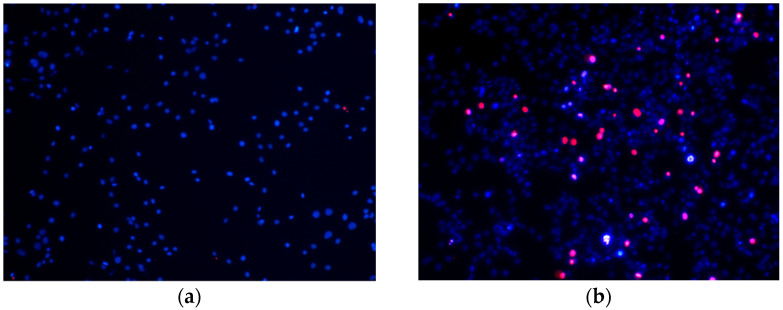
Effectivity of differently prepared extracts from elderflowers in sustaining the viability of glial cells under moderate oxidative stress conditions. Cells were incubated for 6 h with 75 µM H_2_O_2_ or appropriate solvent (control) and additionally in the presence of various concentrations of the elderflower extracts. At the end of experiment, the cells were double-stained with Hoechst 33,342 and PI, and the viability was assessed under fluorescence microscope. Original magnification x20. For details, see the Materials and Methods section. Typical photographs of cells (**a**) Control, (**b**) 75 µM H_2_O_2_, (**c**) cell viability (%) after treatment with H_2_O_2_ in the presence of WES, (**d**) cell viability (%) after treatment with H_2_O_2_ in the presence of Pg-WES, (**e**) cell viability (%) after treatment with H_2_O_2_ in the presence of EES. * statistically significant difference in the number of viable cells vs. solvent-only treated sample; # statistically significant difference in the number of dead cells vs. solvent-only treated sample.

**Table 1 pharmaceutics-14-02831-t001:** The amount of predominant phenolic compounds and the total content of phenolic compounds in investigated extracts.

Extract Type	Dominant Compounds, (μg/mL)					Total Content of Phenolic Compounds (PC), (mg/mL)
	Chlorogenic Acid	Neochlorogenic Acid	Quercetin	Isoquercetin (Quercetin 3-*O*-Glucoside)	Rutin (Quercetin 3-*O*-Rutinoside)	
**WES**	680.4 ± 12.2	54.7 ± 3.2	9.1 ± 0.8	122.8 ± 9.2	680.8 ± 14.8	13.8 ± 1.54
**Pg-WES**	751.2 ± 16.2 *	64.1 ± 4.1	12.8 ± 1.5 *	205.2 ± 7.5 *	991.4 ± 12.9 *	30.3 ± 1.92 *
**EES**	857.5 ± 10.1 *	71.0 ± 3.9 *	9.8 ± 0.9	283.6 ± 8.9 *	1338.3 ± 14.5 *	46.8 ± 2.37 *

* statistically significant difference vs. WES.

**Table 2 pharmaceutics-14-02831-t002:** The stability of investigated extracts.

Extract Type	Storage Period, (Months)	Storage at 20 ± 2 °C		Storage at 6–8 °C	
		Organoleptic Properties	Total Content of Phenolic Compounds, (mg/mL)	Organoleptic Properties	Total Content of Phenolic Compounds, (mg/mL)
**WES**	0	Brown transparent solution	14.2 ± 1.1	Brown transparent solution	14.2 ± 1.1
	1	Brown turbid solution	13.6 ± 0.8	N.ch.	13.8 ± 1.2
	3	Extremely turbid solution	---	Brown turbid solution	12.9 ± 1.2
	6	Extremely turbid solution	---	Brown turbid solution	**11.6 ± 1.3** *
**Pg-WES**	0	Brown transparent solution	31.6 ± 1.9	Brown transparent solution	31.6 ± 1.9
	1	N.ch.	30.4 ± 1.3	N.ch.	29.7 ± 1.5
	3	N.ch.	29.9 ± 1.6	N.ch.	31.0 ± 1.1
	6	Turbid solution	30.8 ± 2.1	N.ch.	30.0 ± 1.6
**EES**	0	Dark brown transparent solution	47.5 ± 2.0	Dark brown transparent solution	47.5 ± 2.0
	1	N.ch.	48.1 ± 1.4	N.ch.	46.8 ± 2.1
	3	N.ch.	46.8 ± 2.2	N.ch.	48.2 ± 1.6
	6	N.ch.	46.5 ± 2.5	N.ch.	46.1 ± 1.8

* statistically significant difference vs. the starting point (0 months); N.ch.—No changes.

## Data Availability

Not applicable.

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
