# Peer review of "Comparison of the Formulation, Stability and Biological Effects of Hydrophilic Extracts from Black Elder Flowers (Sambucus nigra L.)"

_pharmaceutics, 2022, doi:10.3390/pharmaceutics14122831_

Round 1
Reviewer 1 Report
It is really good research and in general the explanation of the paper is coherent. Congratulations to the authors
Line 215-220 why is this happening? More explanation is needed.
More explanation is needed on the stability of the extracts, is there any biochemical explanation when PEG is used, why are the organoleptic properties preserved, what happens? Line 247-249
Author Response
Dear Reviewer,
Thank you very much for the analysis and useful suggestions for the paper.
Point: Line 215-220 why is this happening? More explanation is needed.
Response : The explanation is presented in the Discussion: lines 370-379 and 383-385.
Point: More explanation is needed on the stability of the extracts, is there any biochemical explanation when PEG is used, why are the organoleptic properties preserved, what happens? Line 247-249
Response : The explanation is given in the Discussion: lines 385-394.

Reviewer 2 Report
Dear authors
Regarding the Manuscript "Comparison of the formulation, stability and biological effects of hydrophilic extracts from black elder flowers (Sambucus nigra L.)"
I recommend this article review with "Minor revision".
- Abbreviation not defined in abstract and introduction: Usually, an abbreviation needs to be spelled out once in the Abstract, again in the main text, and used consistently thereafter.
- Introduction should be summarized including many sentences with the same mean.
- Extraction process was done according to Ref.?
- It is preferable to combine the results with the discussion.
- There are many comments in manuscript.
With my best wishes
Mona M. Ismail

Author Response
Dear Reviewer,
Thank you very much for the analysis and useful suggestions for the paper.
Point: Abbreviation not defined in abstract and introduction: Usually, an abbreviation needs to be spelled out once in the Abstract, again in the main text, and used consistently thereafter..
Response : The abbreviations have been defined and revised throughout the manuscript.
Point: Introduction should be summarized including many sentences with the same mean.
Response : It has been revised as suggested.
Point: Extraction process was done according to Ref.?
Response : section “Methods” was supplemented with references
Point: It is preferable to combine the results with the discussion
Response : We would like to follow the recommendations of Pharmaceutics to have Results and Discussion separated.
Point: There are many comments in manuscript.
Response : It has been revised as suggested

Reviewer 3 Report
The paper presents an in-depth analysis of the application of elderberry flower extracts comparing their chemical composition and stability, and the ability to neutralize ROS and to sustain the viability of C6 glial cells under oxidative stress.
1. First and foremost, the authors have carried out an extensive literature review relevant to the study. However, it is recommended that the authors incorporate an analysis of the ethnopharmacological uses of elderberry flower extracts based on traditional medicinal applications.
2. Table 2 is presented in a detailed manner, albeit the interpretation presented in the Results pertaining to this table does not entirely tally with the results, and therefore, requires revision.
3. Images (a) to (e) presented in Figure 3 are somewhat unclear. It is recommended that the authors use clearer images.
Author Response
Dear Reviewer,
Thank you very much for the analysis and useful suggestions for the paper.
Point: First and foremost, the authors have carried out an extensive literature review relevant to the study. However, it is recommended that the authors incorporate an analysis of the ethnopharmacological uses of elderberry flower extracts based on traditional medicinal applications.
Response : Application of elder flower extract in traditional medicine was added as recommended (introduction, lines 53-55)
Point: Table 2 is presented in a detailed manner, albeit the interpretation presented in the Results pertaining to this table does not entirely tally with the results, and therefore, requires revision.
Response: We have addressed this issue in the Discussion.
Point: Images (a) to (e) presented in Figure 3 are somewhat unclear. It is recommended that the authors use clearer images.
Response: We have updated the figure.
